# GEOMETRY-CONSISTENT NEURAL SHAPE REPRESENTATION WITH IMPLICIT DISPLACEMENT FIELDS

**Wang Yifan**
ETH Zurich
ywang@inf.ethz.ch

**Lukas Rahmann**
ETH Zurich
lukas.rahmann@gmail.com

**Olga Sorkine-Hornung**
ETH Zurich
sorkine@inf.ethz.ch

## ABSTRACT

We present *implicit displacement fields*, a novel representation for detailed 3D geometry. Inspired by a classic surface deformation technique, displacement mapping, our method represents a complex surface as a smooth base surface plus a displacement along the base's normal directions, resulting in a frequency-based shape decomposition, where the high-frequency signal is constrained geometrically by the low-frequency signal. Importantly, this disentanglement is unsupervised thanks to a tailored architectural design that has an innate frequency hierarchy by construction. We explore implicit displacement field surface reconstruction and detail transfer and demonstrate superior representational power, training stability, and generalizability. Code and data available at: https://github.com/yifita/idf

## 1 INTRODUCTION

Neural implicit functions have emerged as a powerful tool for representing a variety of signals. Compared to conventional discrete representations, neural implicits are continuous and thus not tied to a specific resolution. Recently, neural implicits have gained significant attraction in a variety of applications ranging from 3D reconstruction (Sitzmann et al., 2019; Niemeyer et al., 2020; Yariv et al., 2020; Peng et al., 2020), neural rendering (Mildenhall et al., 2020; Pumarola et al., 2020), image translation (Skorokhodov et al., 2020; Chen et al., 2020a) to deformation approximation (Deng et al., 2019). In this paper, we focus on neural implicit representations for 3D geometry.

While neural implicits can theoretically model geometry with infinite resolution, in practice the output resolution is dependent on the representational power of neural nets. So far, the research community approaches the problem from two main directions. The first is to partition the implicit function using spatial structures (Chabra et al., 2020; Jiang et al., 2020; Liu et al., 2020; Takikawa et al., 2021), thus making the memory and computation demands dependent on the geometric complexity. The other direction focuses on improving networks' ability to represent high-frequency signals, either in a preprocessing step (referred to as positional encoding) (Mildenhall et al., 2020) or by using sinusoidal representation networks (SIREN) (Sitzmann et al., 2020). However, training these networks is very challenging, as they are prone to overfitting and optimization local minima.

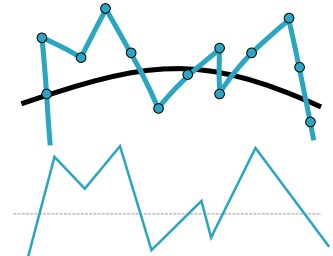

Inspired by the classic computer graphics technique, displacement mapping (Cook, 1984; Cook et al., 1987), we propose a novel parameterization of neural implicit functions, *implicit displacement field*, abbreviated as IDF, to circumvent the above issues. Our method automatically disentangles a given detailed shape into a coarse base shape represented as a continuous, low-frequency signed distance function and a continuous high-frequency implicit displacement field, which offsets the base iso-surface along the normal direction.

Figure 1: Displacement mapping in 1D. The detailed surface (upper blue) is created by offsetting samples of the base surface (upper black) using the height map shown below.

The key novelty of our approach lies in extending the classic displacement mapping, which is discrete and lies only on the base surface, to a continuous function in the $\mathbb{R}^3$ domain and incorporating it into contemporary neural implicit representations, ergo achieving a disentanglement of geometric details in an unsupervised manner.

Our main technical contribution includes

1. a principled and theoretically grounded extension of explicit discrete displacement mapping to the implicit formulation,

2. a neural architecture that creates a geometrically interpretable frequency hierarchy in the neural implicit shape representation by exploiting the inductive bias of SIRENs, and

3. introducing transferable implicit displacement fields by replacing the common coordinates input with carefully constructed transferrable features, thus opening up new opportunities for implicit geometry manipulation and shape modeling.

Systematic evaluations show that our approach is significantly more powerful in representing geometric details, while being lightweight and highly stable in training.

## 2 RELATED WORK

**Hierachical neural implicit shape representation.** Neural implicit shape representation was initially proposed by several works concurrently (Park et al., 2019; Chen & Zhang, 2019; Mescheder et al., 2019), and since then many works have sought to introduce hierarchical structures into the neural representation for better expressiveness and generalizability. The majority of these methods focus on spatial structures. Chabra et al. (2020); Saito et al. (2019; 2020) use sparse regular voxels and dense 2D grid, respectively, to improve detail reconstruction. In the spirit of classic approaches, *e.g.* Frisken et al. (2000); Ohtake et al. (2003), Liu et al. (2020); Takikawa et al. (2021); Martel et al. (2021) store learned latent codes in shape-adaptive octrees, leading to significantly higher reconstruction quality and increased rendering speed. A common disadvantage of these methods is that the memory use and model complexity are directly tied to the desired geometric resolution. In parallel, other proposed methods learn the spatial partition. Some of these methods decompose the given shape using parameterized templates, such as anisotropic Gaussians (Genova et al., 2019), convex shape CVXNet (Deng et al., 2020; Chen et al., 2020b) or simple primitives (Hao et al., 2020), while others represent local shapes with small neural networks and combine them together either using Gaussians (Genova et al., 2020) or surface patches (Tretschk et al., 2020). Due to limitations of template functions and delicate spatial blending issues, these methods can only handle very coarse geometries.

Concurrently, Li & Zhang (2021) propose a two-level neural signed distance function for single-view reconstruction. Exploiting the fact that most man-made shapes have flat surfaces, it represents a given shape as a coarse SDF plus a frontal and rear implicit displacement map for better detail construction. Besides having entirely different applications – we focus on representing significantly higher geometry resolutions – our implicit displacement is grounded in geometry principles and applies to general shapes.

**High-frequency representation in neural networks** As formally explained by Xu (2018); Xu et al. (2019); Rahaman et al. (2019); Basri et al. (2020), neural networks have a tendency to learn low-frequency functions. To combat this issue, Mildenhall et al. (2020) incorporate "positional encoding" for neural rendering and demonstrate remarkable progress in terms of detail reconstruction, which is a sinusoidal mapping for the input signal, a practice later theoretically justified by Tancik et al. (2020). Alternatively, SIREN also shows impressive advances in detail representation by replacing ReLU activation with sin functions. With these new networks gaining popularity, a few works delve deeper and apply a coarse-to-fine frequency hierarchy in the training process for deformable shape representation (Park et al., 2020b) and meshing (Hertz et al., 2021). In our method, we also create a frequency hierarchy by leveraging this new form of networks – not only in the training scheme but also explicitly in the construction of the networks to reflect our geometry-motivated design principles.

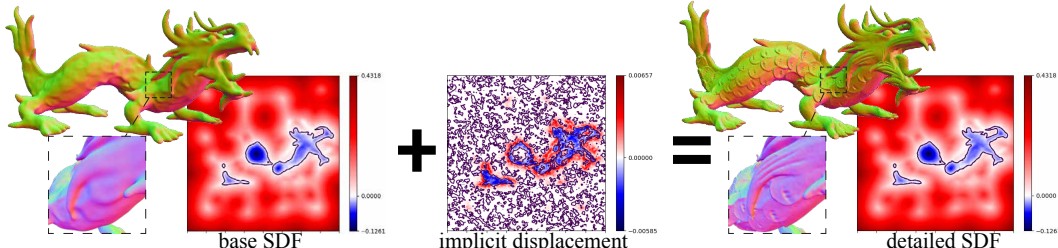

base SDF      implicit displacement      detailed SDF

Figure 2: Method overview. We represent detailed geometries as a sum of a coarse base shape represented as low-frequency signed distance function and a high-frequency implicit displacement field, which offsets the base iso-surface along the base's normal directions.

**Detail transfer**    Detail transfer refers to transplanting the disentangled geometric details from a source shape onto a target object with high fidelity and plausibility. Classic detail transfer methods represent surface details as normal displacements (Botsch et al., 2010; Zhou et al., 2007; Sorkine & Botsch, 2009). The majority of them are parametric (Ying et al., 2001; Biermann et al., 2002; Sorkine et al., 2004; Zhou et al., 2006; Takayama et al., 2011), relying on a consistent surface parameterization between the source and the target shape. Non-parametric approaches (Chen et al., 2012; Berkiten et al., 2017), on the other hand, find best-matching surface patches between the source and target, and copy the details iteratively from coarse to fine. These classic approaches produce high quality results, but often require a pre-defined base surface or abundant user inputs. In the "deep" realm, DeepCage (Yifan et al., 2020) proposed a neural deformation method that maps solely the coarse geometry, hence allowing detail transfer without tackling detail disentanglement. Hertz et al. (2020) learn the coarse-to-detail correspondence iteratively from multi-scale training data, while Chen et al. (2021) synthesizes details by upsampling a coarse voxel shape according to a style code of another shape using GANs. All of these approaches use explicit representations, hence they are subject to self-intersection and resolution limitations. $D^2IM$-Net (Li & Zhang, 2021) uses two planar displacement maps to transfer surface details by mapping the coordinates of the source and target shapes using part segmentation, thus limiting the application to man-made rigid shapes. In comparison, our method does not require any correspondence mapping.

# 3 METHOD

We represent a shape with fine geometric details using two SIREN networks of different frequencies in the activation functions. The SIREN with lower frequency describes a smooth base surface; the SIREN with higher frequency adds microstructure to the base iso-surface by producing an implicit displacement field along the base's normal direction (see Figure 2).

In this section, we first formally define implicit displacement field by generalizing the classic explicit and discrete displacement mapping in Sec 3.1, then in Sec 3.2 we introduce the network architectures and training strategies that are tailored to this definition, finaly in Sec 3.3 we extend the implicit displacement to address transferability.

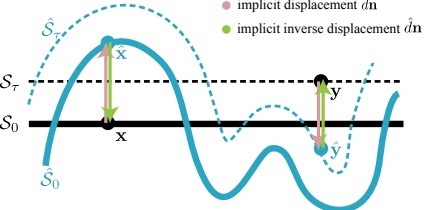

Figure 3: An implicit displacement field for a 1D-curve. The displacement is defined not only on the zero-isosurface $S_0$ but also on arbitrary isosurfaces $S_\tau$

## 3.1 IMPLICIT DISPLACEMENT FIELDS

In classic displacement mapping as shown in Figure 1, high-frequency geometric details are obtained on a smooth base surface by taking samples from the base surface and offsetting them along their normal directions by a distance obtained (with interpolation) from a discrete height map. Two elements in this setting impede a direct adaptation for implicit shape representation: 1. the displacement mapping is defined discretely and only on the base surface, whereas implicit surface functions are typically defined continuously on the $\mathbb{R}^3$ domain; 2. the base surface is known and fixed, whereas our goal is to learn the base surface and the displacement jointly on-the-fly.

Addressing the above challenges, we first define *implicit displacement fields* (IDF), which are continuous analog to height maps that extend displacement mapping to the $\mathbb{R}^3$ domain.

**Definition 1.** *Given two signed distance functions $f$ and $\hat{f}$ and their respective iso-surfaces at a given value $\tau \in \mathbb{R}$, $\mathcal{S}_\tau = \left\{ \mathbf{x} \in \mathbb{R}^3 | f(\mathbf{x}) = \tau \right\}$ and $\hat{\mathcal{S}}_\tau = \left\{ \mathbf{x} \in \mathbb{R}^3 | \hat{f}(\mathbf{x}) = \tau \right\}$, an* implicit displacement field *$d: \mathbb{R}^3 \rightarrow \mathbb{R}$ defines the deformation from $\mathcal{S}_\tau$ to $\hat{\mathcal{S}}_\tau$ such that*

$$f(\mathbf{x}) = \hat{f}(\mathbf{x} + d(\mathbf{x})\mathbf{n}), \text{ where } \mathbf{n} = \frac{\nabla f(\mathbf{x})}{\|\nabla f(\mathbf{x})\|}. \tag{1}$$

This definition is schematically illustrated in Figure 3, where the iso-surface $\mathcal{S}_0$ and $\mathcal{S}_\tau$ are mapped to $\hat{\mathcal{S}}_0$ and $\hat{\mathcal{S}}_\tau$ with the same implicit displacement field $d$. Notably, the height map in classic displacement mapping is a discrete sampling of IDF for the limited case $\tau = 0$.

In the context of surface decomposition, our goal is to estimate the base surface $f$ and the displacement $d$ given an explicitly represented detailed surface $\hat{\mathcal{S}}_0$. Following equation 1, we can do so by minimizing the difference between the base and the ground truth signed distance at query points $\mathbf{x} \in \mathbb{R}^3$ and their displaced position $\hat{\mathbf{x}} = \mathbf{x} + d(\mathbf{x})\mathbf{n}$, *i.e.*, $\min |f(\mathbf{x}) - \hat{f}_{\text{GT}}(\hat{\mathbf{x}})|$.

However, this solution requires evaluating $\hat{f}_{\text{GT}}(\hat{\mathbf{x}})$ dynamically at variable positions $\hat{\mathbf{x}}$, which is a costly operation as the detailed shapes are typically given in explicit form, *e.g.*, as point clouds or meshes. Hence, we consider the inverse implicit displacement field $\hat{d}$, which defines a mapping from $\hat{\mathcal{S}}_\tau$ to $\mathcal{S}_\tau$, $f\left(\hat{\mathbf{x}} + \hat{d}(\hat{\mathbf{x}})\mathbf{n}\right) = \hat{f}(\hat{\mathbf{x}})$, as depicted in Figure 3.

Assuming the displacement distance is small, we can approximate $\mathbf{n}$, the normal after inverse displacement, with that before the inverse displacement, *i.e.*

$$f\left(\hat{\mathbf{x}} + \hat{d}(\hat{\mathbf{x}})\hat{\mathbf{n}}\right) = \hat{f}(\hat{\mathbf{x}}), \text{ where } \hat{\mathbf{n}} = \frac{\nabla f(\hat{\mathbf{x}})}{\|\nabla f(\hat{\mathbf{x}})\|}. \tag{2}$$

This is justified by the following theorem and corollary, which we prove in the Appendix A.

**Theorem 1.** *If function $f : \mathbb{R}^n \rightarrow \mathbb{R}$ is differentiable, Lipschitz-continuous with constant $L$ and Lipschitz-smooth with constant $M$, then $\|\nabla f(\mathbf{x} + \delta \nabla f(\mathbf{x})) - \nabla f(\mathbf{x})\| \leq |\delta| LM$.*

**Corollary 1.** *If a signed distance function $f$ satisfying the eikonal equation up to error $\epsilon > 0$, $|\|\nabla f\| - 1| < \epsilon$, is Lipschitz-smooth with constant $M$, then $\|\nabla f(\mathbf{x} + \delta \nabla f(\mathbf{x})) - \nabla f(\mathbf{x})\| < (1 + \epsilon)|\delta|M$.*

Given $\mathbf{n} = \frac{\nabla f(\mathbf{x})}{\|\nabla f(\mathbf{x})\|}$, $\hat{\mathbf{n}} = \frac{\nabla f(\hat{\mathbf{x}})}{\|\nabla f(\hat{\mathbf{x}})\|}$, and $\hat{\mathbf{x}} = \mathbf{x} + d(\mathbf{x})\mathbf{n}$, let $\delta = \frac{d(\mathbf{x})}{\|\nabla f(\mathbf{x})\|}$, we can show $\|\hat{\mathbf{n}} - \mathbf{n}\| \leq \frac{1 + \epsilon}{1 - \epsilon} |\delta| M$ (*c.f.* Appendix A). In other words, the difference of $\hat{\mathbf{n}}$ and $\mathbf{n}$ is bounded by a small constant. Thus we obtain the approximation in equation 2, which allows us to presample training samples $\{\hat{\mathbf{x}}\}$ and use precomputed $\hat{f}_{\text{GT}}(\hat{\mathbf{x}})$ or its derivatives (see Sec 3.2) for supervision.

## 3.2 NETWORK DESIGN AND TRAINING

The formulation of (inverse) implicit field in the previous section is based on three assumptions: (i) $f$ is smooth, (ii) $d$ is small, (iii) $f$ satisfies the eikonal constraint up to an error bound. In this section, we describe our network architecture and training technique, with emphasis on meeting these requirements.

**Network architecture.** We propose to model $f$ and $\hat{d}$ with two SIRENs denoted as $\mathcal{N}^{\omega_B}$ and $\mathcal{N}^{\omega_D}$, where $\omega_B$ and $\omega_D$ refer to the frequency hyperparameter in the sine activation functions $\mathbf{x} \mapsto \sin(\omega \mathbf{x})$. Evidently, as shown in Figure 4, $\omega$ dictates an upper bound on the frequencies the network is capable of representing, thereby it also determines the network's inductive bias for smoothness. Correspondingly, we enforce the smoothness of $f$ and detail-representing capacity of $\hat{d}$ using a smaller $\omega_B$ and a larger $\omega_D$, *e.g.* $\omega_B = 15$ and $\omega_D = 60$. Moreover, we add a scaled $\tanh$ activation to the last linear layer of $\mathcal{N}^{\omega_D}$, *i.e.* $\alpha \tanh(\cdot)$, which ensures that the displacement distance is smaller than the constant $\alpha$. More insight about the choice of $\omega_{B/D}$ is detailed in Section B.2.

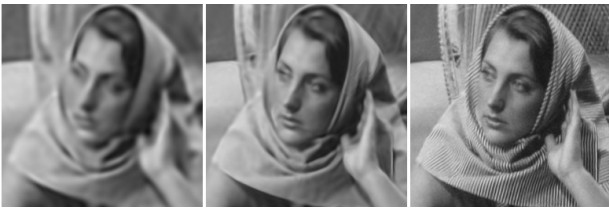
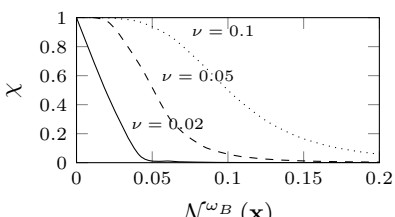

Figure 4: Smoothness control via SIREN's frequency hyperparameter $\omega$. Overfitting SIREN to the right image with $\omega = 30$ (first) and $\omega = 60$ (middle) shows that smaller $\omega$ leads to a smoother result.

Figure 5: Attenuation as a function of base SDF.

Networks containing high-frequency signals, *e.g.* $\mathcal{N}^{\omega_D}$, require large amounts of accurate ground truth data for supervision to avoid running into optimization local minima (Park et al., 2020b). Consequently, when dense and accurate ground truth SDF values are not available, high-frequency signals often create artifacts. This is often the case in void regions when learning from point clouds, as only implicit regularization and fuzzy supervision is applied (see the first and last terms of equation 4). Hence, we apply an attenuation function $\chi\left(\mathcal{N}^{\omega_B}\right) = \frac{1}{1+\left(\mathcal{N}^{\omega_B}(\mathbf{x})/\nu\right)^4}$ to subdue $\mathcal{N}^{\omega_D}$ far from the base surface, where $\nu$ determines the speed of attenuation as depicted in Figure 5.

Combining the aforementioned components, we can compute the signed distance of the detailed shape at query point $\mathbf{x}$ in two steps:

$$f\left(\mathbf{x}\right) = \mathcal{N}^{\omega_B}\left(\mathbf{x}\right), \quad \hat{f}\left(\mathbf{x}\right) = \mathcal{N}^{\omega_B}\left(\mathbf{x} + \chi\left(f\left(\mathbf{x}\right)\right)\mathcal{N}^{\omega_D}\left(\mathbf{x}\right)\frac{\nabla f\left(\mathbf{x}\right)}{\|\nabla f\left(\mathbf{x}\right)\|}\right). \qquad (3)$$

**Training.** We adopt the loss from SIREN, which is constructed to learn SDFs directly from oriented point clouds by solving the eikonal equation with boundary constraint at the on-surface points. Denoting the input domain as $\Omega$ (by default set to $[-1, 1]^3$) and the ground truth point cloud as $\mathcal{P} = \{(\mathbf{p}_i, \mathbf{n}_i)\}$, the loss computed as in equation 4, where $\lambda_{\{0,1,2,3\}}$ denote loss weights:

$$\mathcal{L}_{\hat{f}} = \sum_{\mathbf{x}\in\Omega}\lambda_0\left|\|\nabla\hat{f}\left(\mathbf{x}\right)\| - 1\right| + \sum_{(\mathbf{p},\mathbf{n})\in\mathcal{P}}\left(\lambda_1|\hat{f}\left(\mathbf{p}\right)| + \lambda_2\left(1 - \left\langle\nabla\hat{f}\left(\mathbf{p}\right), \mathbf{n}\right\rangle\right)\right)$$
$$+ \sum_{\mathbf{x}\in\Omega\setminus\mathcal{P}}\lambda_3\exp\left(-100\,\hat{f}\left(\mathbf{x}\right)\right). \qquad (4)$$

As the displacement and the attenuation functions depend on the base network, it is beneficial to have a well-behaving base network when training the displacement (see Sec 4.2). Therefore, we adopt a progressive learning scheme, which first trains $\mathcal{N}^{\omega_B}$, and then gradually increase the impact of $\mathcal{N}^{\omega_D}$. Notably, similar frequency-based coarse-to-fine training techniques are shown to improve the optimization result in recent works (Park et al., 2020b; Hertz et al., 2021).

We implement the progressive training via symmetrically diminishing/increasing learning rates and loss weights for the base/displacement networks. For brevity, we describe the procedure for loss weights only, and we apply the same to the learning rates in our implementation. First, we train $\mathcal{N}^{\omega_B}$ by substituting $\hat{f}$ in the loss equation 4 with $f$, resulting a base-only loss denoted $\mathcal{L}_f$. Then, starting from a training percentile $T_m \in [0, 1]$, we combine $\mathcal{L}_f$ and $\mathcal{L}_{\hat{f}}$ via $\kappa\mathcal{L}_f + (1 - \kappa)\mathcal{L}_{\hat{f}}$ with $\kappa = \frac{1}{2}\left(1 + \cos\left(\pi\frac{(t-T_m)}{(1-T_m)}\right)\right)$, where $t \in [T_m, 1]$ denotes the current training progress.

### 3.3 TRANSFERABLE IMPLICIT DISPLACEMENT FIELD.

In classic displacement mapping, the displacement is queried by the UV-coordinates from surface parameterization, which makes the displacement independent of deformations of the base surface. We can achieve similar effect *without parameterization* by learning *query features*, which emulate the UV-coordinates to describe the location of the 3D query points *w.r.t.* the base surface.

We construct the query features using two pieces of information: (i) a global context descriptor, $\phi(\mathbf{x})$, describing the location of the query point in relation to the base surface in a semantically

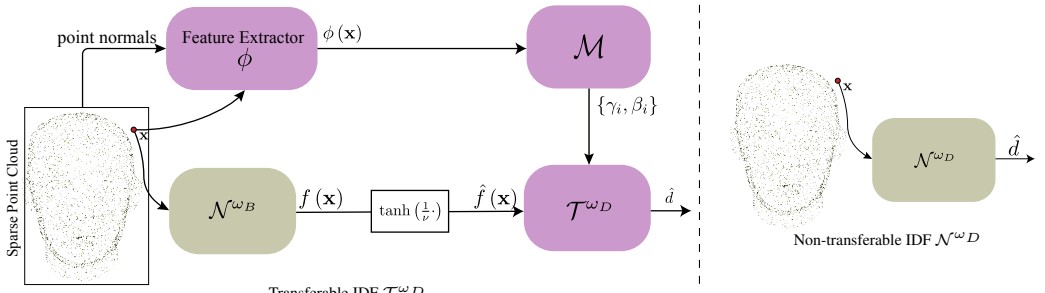

Figure 6: Illustrations for transferable and non-transferable implicit fields. The transferable modules are in pink and the shape-specific modules are in yellow. Instead of consuming the euclidean coordinates, the transferable displacement network takes a scale-and-translation invariant feature as inputs, which describes the relative position of the query point to the base shape.

meaningful way, (ii) the base signed distance value $f(\mathbf{x})$, which gives more precise relative location with respect to the base surface. Since both are differentiable *w.r.t.* the euclidean coordinates of the query point, we can still train $\mathcal{N}^{\omega_D}$ using derivatives as in equation 4.

Our global context descriptor is inspired by Convolutional Occupancy Networks (Peng et al., 2020). Specifically, we project the sparse on-surface point features obtained using a conventional point cloud encoder onto a regular 3D (or 2D, *c.f.* Sec 4.3) grid, then use a convolutional module to propagate sparse on-surface point features to the off-surface area, finally obtain the query feature using bilinear interpolation. We use normals instead of point positions as the input to the point cloud encoder, making the features scale-invariant and translation-invariant. Note that ideally the features should also be rotation-invariant. Nevertheless, as we empirically show later, normal features can in fact generalize under small local rotational deformations, which is sufficient for transferring displacements between two roughly aligned shapes. We leave further explorations in this direction for future work.

$\mathcal{N}^{\omega_D}$ is tasked to predict the displacement conditioning on $\phi(\mathbf{x})$ and $f(\mathbf{x})$. However, empirical studies (Chan et al., 2020) suggest that SIREN does not handle high-dimensional inputs well. Hence, we adopt the FiLM conditioning (Perez et al., 2018; Dumoulin et al., 2018) as suggested by Chan et al. (2020), which feeds the conditioning latent vector as an affine transformation to the features of each layer. Specifically, a mapping network $\mathcal{M}$ converts $\phi(\mathbf{x})$ to a set of C-dimensional frequency modulators and phase shifters $\{\boldsymbol{\gamma}_i, \boldsymbol{\beta}_i\}$, which transform the $i$-th linear layer to $\left(\mathbf{1} + \frac{1}{2}\boldsymbol{\gamma}_i\right) \circ (\mathbf{W}_i \mathbf{x} + \mathbf{b}_i) + \boldsymbol{\beta}_i$, where $\mathbf{W}_i$ and $\mathbf{b}_i$ are the parameters in the linear layer and $\circ$ denotes element-wise multiplication. Finally, since SIREN assumes inputs in range $(-1, 1)$, we scale $f$ using $\bar{f}(\mathbf{x}) = \tanh\left(\frac{1}{\nu} f(\mathbf{x})\right)$ to capture the variation close to the surface area, where $\nu$ is the attenuation parameter described in section 3.2.

Figure 6 summarizes the difference between transferable and non-transferable displacement fields. Formally, the signed distance function of the detailed shape in equation 3 can be rewritten as

$$\hat{f}(\mathbf{x}) = \mathcal{N}^{\omega_B}\left(\mathbf{x} + \chi\left(f(\mathbf{x})\right) \mathcal{T}^{\omega_D}\left(\bar{f}(\mathbf{x}), \mathcal{M}\left(\phi(\mathbf{x})\right)\right) \frac{\nabla f(\mathbf{x})}{\|\nabla f(\mathbf{x})\|}\right). \tag{5}$$

## 4   RESULTS

We now present the results of our method. In Sec 4.1, we evaluate our networks in terms of geometric detail representation by comparing with state-of-the-art methods on the single shape fitting task. We then evaluate various design components in an ablation study in Sec 4.2. Finally, we validate the transferability of the displacement fields in a detail transfer task in Sec 4.3. Extended qualitative and quantitative evaluations are included in section B.

**Implementation details.**   By default, both the base and the displacement nets have 4 hidden layers with 256 channels each. The maximal displacement $\alpha$, attenuation factors $\nu$, and the switching training percentile is set to $T_m$ are set to $0.05$, $0.02$ and $0.2$ respectively; The loss weights $\lambda_{\{0,1,2,3\}}$ in equation 4 are set to 5, 400, 40 and 50. We train our models for 120 epochs using ADAM opti-

Chamfer distance points to point distance $\cdot 10^{-3}$ / normal cosine distance $\cdot 10^{-2}$

| | Progressive FFN | NGLOD (LOD4) | NGLOD (LOD6) | SIREN-3 $\omega = 60$ | SIREN-7 $\omega = 30$ | SIREN-7 $\omega = 60$ | Direct Residual | D-SDF | Ours |
|---|---|---|---|---|---|---|---|---|---|
| SketchFab-16 | 5.47/3.77 | 2.27/4.24 | 1.35/1.97 | 9.85/6.64 | 4.85/2.56 | - | 181/59.0 | 2.85/4.39 | **1.22/1.25** |

Table 1: Quantitative comparison. Among the benchmarking methods, only NGLOD at LOD-6, using $256\times$ number of parameters compared to our model, can yield results close to ours. SIREN models with larger $\omega$ have convergence issues: despite our best efforts, the models still diverged in most cases. Please refer to Table 4 for a more comprehensive evaluation.

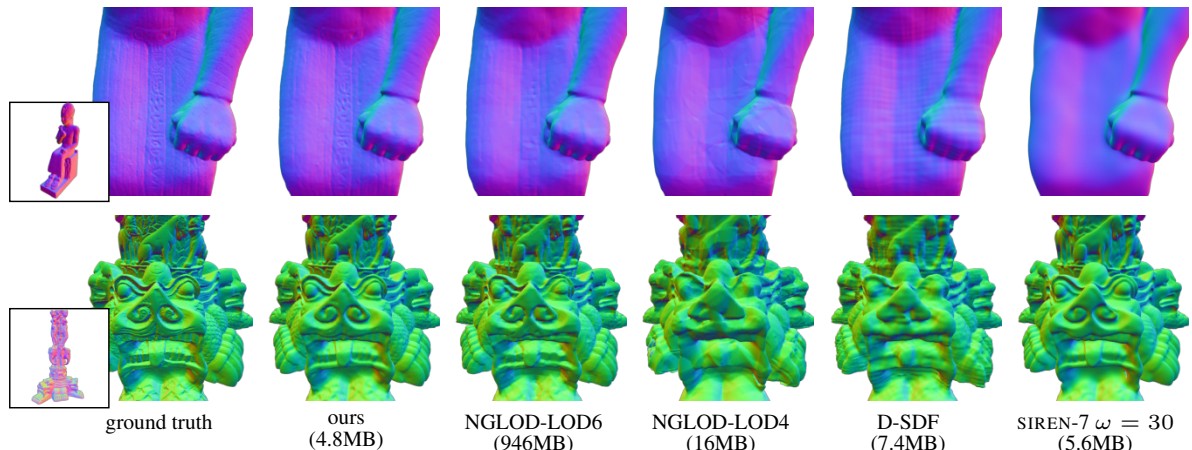

| ground truth | ours (4.8MB) | NGLOD-LOD6 (946MB) | NGLOD-LOD4 (16MB) | D-SDF (7.4MB) | SIREN-7 $\omega = 30$ (5.6MB) |

Figure 7: Comparison of detail reconstruction (better viewed with zoom-in). We show the best 5 methods and their model sizes according to Table 1, more results are provided in section B.1.

mizer with initial learning rate of 0.0001 and decay to 0.00001 using cosine annealing (Loshchilov & Hutter, 2016) after finishing 80% of the training epochs. We presample 4 million surface points per mesh for supervision. Each training iteration uses 4096 subsampled surface points and 4096 off-surface points uniformly sampled from the $[-1, 1]^3$ bounding box. To improve the convergence rate, we initialize SIREN models by pre-training the base model $\mathcal{N}^{\omega_B}$ (and baseline SIREN, *c.f.* Sec 4.1) to a sphere with radius $0.5$. This initialization is optional for our training but is critical for baseline SIRENs.

**Data.** We test our method using 16 high-resolution shapes, including 14 from Sketchfab (ske, 2021) and 2 from Stanford 3DScanRepo (sta, 2021). Our transferable displacement model is tested using shapes provided by Berkiten et al. (2017), Yang et al. (2020), and Zhou & Jacobson (2016).

## 4.1 DETAIL REPRESENTATION.

We compare our approaches with 5 baseline methods. 1) FFN (Tancik et al., 2020) with SOFTPLUS activation and 8 frequency bands progressively trained from coarse-to-fine, where a total of 8 hidden layers each of size 256 are used to match our model size; additionally we apply a skip-connection in the middle layer as proposed in DeepSDF Park et al. (2019). 2) NGLOD (Takikawa et al., 2021) trained using 4 and 6 levels of detail (LODs) corresponding to $64^3$ and $256^3$ spatial resolution respectively, with LOD4 comparable with our model in terms of the number of parameters. 3) baseline SIREN, for which we trained three different variations in hope of overcoming training divergence issues;. 4) direct residual, where we compose the signed distance value simply as the sum of base and displacement nets, *i.e.* $\hat{f}(\mathbf{x}) = \mathcal{N}^{\omega_B}(\mathbf{x}) + \mathcal{N}^{\omega_D}(\mathbf{x})$. 5) D-SDF (inspired by Pumarola et al. (2021); Park et al. (2020a)), which represents the displacement as a $\mathbb{R}^3$ vector, *i.e.* $\hat{f} = f(\mathbf{x} + \Delta)$, where $\Delta \in \mathbb{R}^3$ is predicted in the second network. We follow network specs of D-Nerf Pumarola et al. (2020), which contains two 8-layer MLP networks with RELU activation and positional encodings. Among these, NGLOD, direct residual and D-SDF requires ground truth SDF for supervision, the rest are trained using our training loss. Two-way point-to-point distance and normal cosine distance are computed as the evaluation metrics on 5 million points randomly sampled from meshes extracted using marching cubes with $512^3$ resolution.

| | $\alpha$ test (with $\nu = 0.02$) | | | | | $\nu$ test with ($\alpha = 0.05$) | | | | |
|---|---|---|---|---|---|---|---|---|---|---|
| | 0.01 | 0.02 | 0.05 | 0.1 | 0.2 | 0.01 | 0.02 | 0.05 | 0.1 | 0.2 |
| point-to-point distance $(\cdot 10^{-3})$ | 1.178 | 1.171 | 1.147 | 1.146 | 1.149 | 1.146 | 1.147 | 1.147 | 1.149 | 1.152 |
| normal cosine distance $(\cdot 10^{-2})$ | 1.525 | 1.490 | 1.252 | 1.251 | 1.260 | 1.254 | 1.253 | 1.251 | 1.250 | 1.274 |

Table 3: Study of the hyperparameters $\alpha$ (left) and $\nu$ (left). The reconstruction accuracy remains stable and highly competetive throughout hyperparameters variation.

As shown in Table 1 and Figure 7, our method outperforms the baseline methods with much higher reconstruction fidelity. NGLOD with 6 LODs is the only method onpar with ours in terms of detail representation, however it requires storing more than 300 times as many as parameters as our model. SIREN networks with larger $\omega$ have severe convergence issues even with sphere initialization (*c.f.* Implementation Details) and gradient clipping. Direct residual doesn't enforce displacement directions and produces large structural artifacts. D-SDF yields qualitatively poor results, as the displacement net is unable to learn meaningful information (more analysis is shown in section B.1.2).

## 4.2 ABLATION STUDY

We study the contributions of different design components, namely the displacement scaling $\alpha \tanh$, the attenuation function $\chi$ and the progressive training.

As Table 2 shows, all the test modes converge within comparable range, even for the model with the least constraints. This shows that our model is robust against violations of theoretical assumptions specified in Sec 3.1. At the same time, the performance rises with increasingly constrained architecture and progressive training, suggesting that the proposed mechanisms further boost training stability.

| $\alpha \tanh$ | $\chi$ | prog. training | average CD $\cdot 10^{-3}$ |
|---|---|---|---|
| | | | 1.44 |
| ✓ | | | 1.41 |
| ✓ | ✓ | | 1.38 |
| ✓ | ✓ | ✓ | 1.24 |

Table 2: Ablation study. Our model benefits from the proposed architectural and training designs, yet it is also robust against variations.

Table 3 shows that in a reasonable range of $\alpha$ and $\nu$ there is very little variance in reconstruction quality, indicating the robustness *w.r.t.* parameter selection. If $\alpha$ is too small, the displacement may no longer be sufficient to correct the difference between the base and detailed surface, causing the slight increase of chamfer distances in the table for $\alpha = \{0.01, 0.02\}$. When $\nu$ is too large (0.2), *i.e.* the high frequency signal is not suppressed in the void region, which leads to higher chamfer distances due to off-surface high-frequency noise.

## 4.3 TRANSFERABILITY

We apply our method to detail transfer in order to validate the transferability of IDF. Specifically, we want to transfer the displacements learned for a source shape to a different aligned target shape. In the first test scenario, the base shape is provided and lies closely to the ground truth detailed surface. In the second scenario, we are only provided with the detailed shapes and thus need to estimate the base and the displacements jointly. The pipeline consists of the following steps: 1) train $\mathcal{N}^{\omega_B}$ by fitting the source shape (or the source base shape if provided), 2) train $\mathcal{T}^{\omega_D}, \mathcal{M}$ and the query feature extractor $\phi$ jointly by fitting the source shape using equation 5 while keeping $\mathcal{N}^{\omega_B}$ fixed, 3) train $\mathcal{N}^{\omega_B}_{\text{new}}$ by fitting the target shape (or the target base shape if provided), 4) evaluate equation 5 by replacing $\mathcal{N}^{\omega_B}$ with $\mathcal{N}^{\omega_B}_{\text{new}}$. To prevent the base network from learning high-frequency details when the base is unknown, we use $\omega_B = 5$ and three 96-channel hidden layers for $\mathcal{N}^{\omega_B}$.

Example outputs for both scenarios are shown in Figure 10, where the base shapes are provided for the *shorts* model. We use a $32^3$ and a $128^2$ grid (for the frontal view), for the *shorts* and *face* model respectively in $\phi$ to extract the query features. Our displacement fields, learned solely from the source shape, generate plausible details on the unseen target shape. The transferred details contain high-frequency signals (*e.g.* the eyebrows on the face), which is challenging for explicit representations. However, for the second scenario the performance degenerates slightly since the displacement field has to compensate errors stemming specifically from the base SDF.

In additional, we evaluate the design of the transferable IDF model by removing the mapping net and the convolutional context descriptor $\phi$. For the former case, we drop the FiLM conditioning

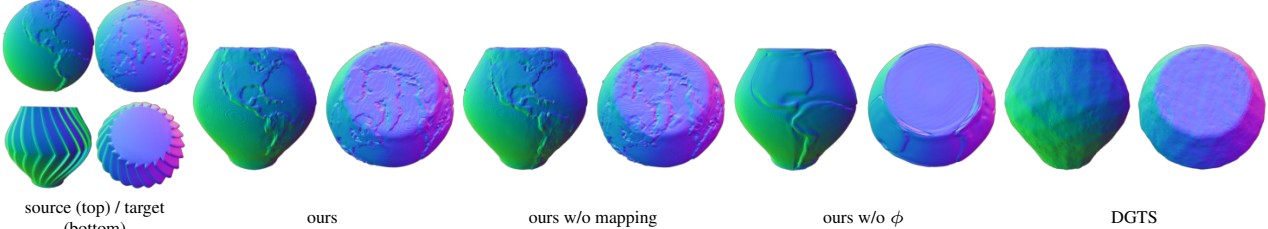

source (top) / target (bottom)  ours  ours w/o mapping  ours w/o $\phi$  DGTS

Figure 9: Transferring spatially-variant geometric details using various methods. Small to severe distortions are introduced when removing different components of the proposed transferable IDF. Thanks to the combination of global/local query feature, our method transfers spatially-variant details while Hertz et al. (2020) can only handle spatially-invariant isometric details.

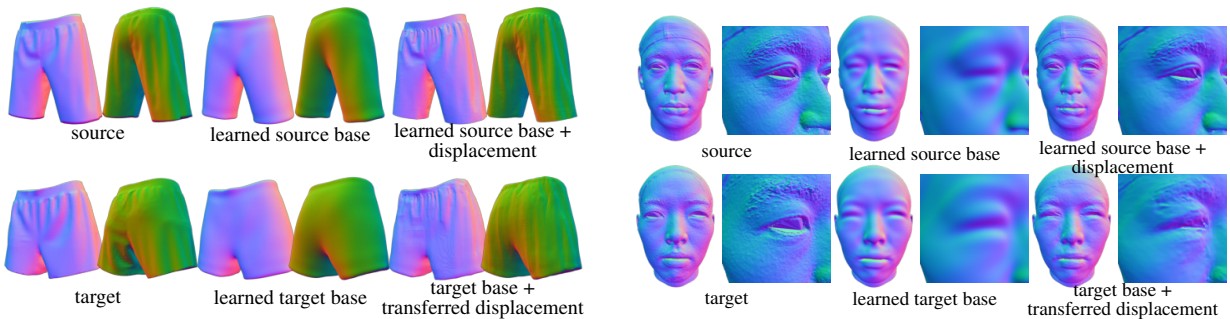

source  learned source base  learned source base + displacement  source  learned source base  learned source base + displacement

target  learned target base  target base + transferred displacement  target  learned target base  target base + transferred displacement

Figure 10: Transferable IDF applied to detail transfer. Left: the base shape is provided and lies closely to the ground truth detailed surface; right: only the detailed shapes are provided, thus the base and the displacements need to estimated jointly.

and simply use concatenation of $\phi(\mathbf{x})$ and $\bar{f}(\mathbf{x})$ as the inputs to $\mathcal{N}^{\omega_D}$; for the latter we directly use the normal at the sampled position as the context descriptor, *i.e.* $\phi(\mathbf{x}) = \nabla f(\mathbf{x})$. As Figure 9 shows, the removal of mapping net and $\phi$ lead to different degrees of feature distortions. We also compare with the DGTS (Hertz et al., 2020), which fails completely at this example since it only consumes local intrinsic features. Furthermore, the effect of scaling $\bar{f}$ is shown in Figure 8, where using unscaled $f$ as input to $\mathcal{T}^{\omega_D}$ leads to artifacts at the boundary.

## 5 CONCLUSION AND LIMITATIONS

In this paper, we proposed a new parameterization of neural implicit functions for detailed geometry representation. Extending displacement mapping, a classic shape modeling technique, our formulation represents a given shape by a smooth base surface and a high-frequency displacement field that offsets the base surface along its normal directions. This resulting frequency partition enables the network to concentrate on regions with rich geometric details, significantly boosting its representational power. Thanks to the theoretically grounded network design, the high-frequency signal is well constrained, and as a result our model shows convergence qualities compared to other models leveraging high-frequency signals, such as SIREN and positional encoding. Furthermore, emulating the deformation-invariant quality of classic displace-

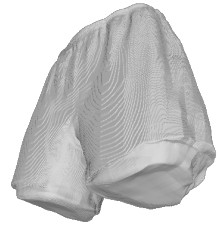

Figure 8: Detail transfer without scaling $\bar{f}$.

ment mapping, we extend our method to enable transferability of the implicit displacements, thus making it possible to use implicit representations for new geometric modeling tasks.

A limitation of our detail transfer application is the necessity to pre-align the two shapes. In future work, we consider exploring sparse correspondences as part of the input, which is a common practice in computer graphics applications, to facilitate subsequent automatic shape alignment.

## ACKNOWLEDGMENTS

This work is sponsored by Apple's AI/ML PhD fellowship program.

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

## A  PROOFS

**Theorem 1.** *If function* $f : \mathbb{R}^n \to \mathbb{R}$ *is differentiable, Lipschitz-continuous with constant* $L$ *and Lipschitz-smooth with constant* $M$, *then* $\|\nabla f\left(\mathbf{x} + \delta\,\nabla f\left(\mathbf{x}\right)\right) - \nabla f\left(\mathbf{x}\right)\| \leq |\delta| L M$.

*Proof.* If a differentiable function $f$ is Lipschitz-continuous with constant $L$ and Lipschitz-smooth with constant $M$, then

$$\|\nabla f\left(\mathbf{x}\right)\| \leq L \tag{6}$$

$$\|\nabla f\left(\mathbf{x}\right) - \nabla f\left(\mathbf{y}\right)\| \leq M\|\mathbf{x} - \mathbf{y}\|. \tag{7}$$

$$\|\nabla f\left(\mathbf{x} + \delta\,\nabla f\left(\mathbf{x}\right)\right) - \nabla f\left(\mathbf{x}\right)\| \leq M\|\delta\,\nabla f\left(\mathbf{x}\right)\| \qquad \text{by equation 7}$$

$$\leq |\delta| L M \qquad \text{by equation 6}$$

$\square$

**Corollary 1.** *If a signed distance function $f$ satisfying the eikonal equation up to error $\epsilon > 0$, $\left|\|\nabla f\| - 1\right| < \epsilon$, is Lipschitz-smooth with constant $M$, then $\|\nabla f(\mathbf{x} + \delta \nabla f(\mathbf{x})) - \nabla f(\mathbf{x})\| < (1 + \epsilon)|\delta|M$.*

*Proof.* $\left|\|\nabla f\| - 1\right| < \epsilon \Rightarrow \|\nabla f\| < \epsilon + 1$. This means $f$ is Lipschitz-continuous with constant $\epsilon + 1$. Then by Theorem 1, $\|\nabla f(\mathbf{x} + \delta \nabla f(\mathbf{x})) - \nabla f(\mathbf{x})\| < |\delta|(1 + \epsilon)M$. $\qquad\square$

Finally we show the upper bound for the normalized gradient, *i.e.*,

$$\|\hat{\mathbf{n}} - \mathbf{n}\| \leq \frac{1 + \epsilon}{1 - \epsilon} |\delta| M, \tag{8}$$

where $\mathbf{n} = \frac{\nabla f(\mathbf{x})}{\|\nabla f(\mathbf{x})\|}$, $\hat{\mathbf{n}} = \frac{\nabla f(\hat{\mathbf{x}})}{\|\nabla f(\hat{\mathbf{x}})\|}$ and $\hat{\mathbf{x}} = \mathbf{x} + d(\mathbf{x})\mathbf{n}$ with $d(\mathbf{x})$ denoting the small displacement.

*Proof.*

$$\|\hat{\mathbf{n}} - \mathbf{n}\| = \left\| \frac{\nabla f(\hat{\mathbf{x}})}{\|\nabla f(\hat{\mathbf{x}})\|} - \frac{\nabla f(\mathbf{x})}{\|\nabla f(\mathbf{x})\|} \right\|. \tag{9}$$

For brevity, we denote $\nabla f(\hat{\mathbf{x}})$ and $\nabla f(\mathbf{x})$ as $\mathbf{u}$ and $\mathbf{v}$. Without loss of generality, we assume $\|\mathbf{u}\| \leq \|\mathbf{v}\|$. Then

$$\|\hat{\mathbf{n}} - \mathbf{n}\| = \left\| \frac{\mathbf{u}}{\|\mathbf{u}\|} - \frac{\mathbf{v}}{\|\mathbf{v}\|} \right\| \tag{10}$$

$$\overset{(*)}{\leq} \frac{1}{\|\mathbf{u}\|} \|\mathbf{u} - \mathbf{v}\| \tag{11}$$

$$\leq \frac{1}{1 - \epsilon} \|\mathbf{u} - \mathbf{v}\| \qquad \text{by Eikonal constraint} \tag{12}$$

$$= \frac{1}{1 - \epsilon} \|\nabla f(\hat{\mathbf{x}}) - \nabla f(\mathbf{x})\| \tag{13}$$

$$= \frac{1}{1 - \epsilon} \left\| \nabla f\left( \mathbf{x} + \frac{d(\mathbf{x})\nabla f(\mathbf{x})}{\|\nabla f(\mathbf{x})\|} \right) - \nabla f(\mathbf{x}) \right\|. \tag{14}$$

Since $|d(\mathbf{x})|$ is a small and $\|\nabla f(\mathbf{x})\|$ is close to 1, we can set $\delta = \frac{d(\mathbf{x})}{\|\nabla f(\mathbf{x})\|}$. Thereby using Corollary 1, we conclude

$$\frac{1}{1 - \epsilon} \left\| \nabla f\left( \mathbf{x} + \frac{d(\mathbf{x})\nabla f(\mathbf{x})}{\|\nabla f(\mathbf{x})\|} \right) - \nabla f(\mathbf{x}) \right\| \leq \frac{1 + \epsilon}{1 - \epsilon} |\delta| M, \tag{15}$$

thus

$$\|\hat{\mathbf{n}} - \mathbf{n}\| \leq \frac{1 + \epsilon}{1 - \epsilon} |\delta| M. \tag{16}$$

Eq. equation 11 can be proved as follows

$$\left\| \overbrace{\frac{\mathbf{u}}{\|\mathbf{u}\|} - \frac{\mathbf{v}}{\|\mathbf{v}\|}}^{\mathbf{d}} \right\| = \left\| \overbrace{\frac{\mathbf{u}}{\|\mathbf{u}\|} - \frac{\mathbf{v}}{\|\mathbf{u}\|}}^{\mathbf{d}'} + (\overbrace{\frac{\mathbf{v}}{\|\mathbf{u}\|} - \frac{\mathbf{v}}{\|\mathbf{v}\|}}^{\mathbf{e}}) \right\|, \tag{17}$$

which depicts the distance of the unit sphere projections of $\mathbf{u}$ and $\mathbf{v}$. Obviously, as shown in Figure 11, $\|\mathbf{d}\| \leq \|\mathbf{d}'\|$ if $\sphericalangle(\mathbf{d}, \mathbf{e}) \geq 90°$.

Since $\mathbf{e} = (\frac{1}{\|\mathbf{u}\|} - \frac{1}{\|\mathbf{v}\|})\mathbf{v}$ and $(\frac{1}{\|\mathbf{u}\|} - \frac{1}{\|\mathbf{v}\|}) \geq 0)$, to show that $\sphericalangle\langle\mathbf{d}, \mathbf{e}\rangle \geq 90°$ is the same as to show that $\sphericalangle\langle\mathbf{d}, \mathbf{v}\rangle \geq 90°$. Indeed:

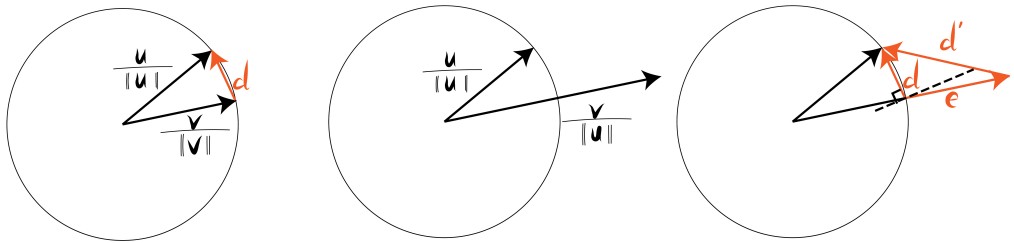

Figure 11: Sketch for proof.

$$\langle \mathbf{d}, \mathbf{v} \rangle = \left\langle \frac{\mathbf{u}}{\|\mathbf{u}\|} - \frac{\mathbf{v}}{\|\mathbf{v}\|}, \mathbf{v} \right\rangle \tag{18}$$

$$= \frac{\langle \mathbf{u}, \mathbf{v} \rangle}{\|\mathbf{u}\|} - \|\mathbf{v}\| \tag{19}$$

$$\leq \frac{\|\mathbf{u}\|\|\mathbf{v}\|}{\|\mathbf{u}\|} - \|\mathbf{v}\| \qquad \text{by Cauchy-Schwarz inequality} \tag{20}$$

$$= 0 \tag{21}$$

$\square$

## B  ADDITIONAL EXPERIMENTS AND RESULTS.

### B.1  ADDITIONAL INFORMATION TO THE COMPARISON IN SEC 4.1

#### B.1.1  ADDITIONAL EVALUATION RESULTS

Below we show the per-model Chamfer distances in addition to the average shown in Table 1. More qualitative results are shown in Figure 16.

Chamfer distance points to point distance $\cdot 10^{-3}$ / normal cosine distance $\cdot 10^{-2}$

| model | Progressive FFN | NGLOD (LOD4) | NGLOD (LOD6) | SIREN-3 $\omega=60$ | SIREN-7 $\omega=30$ | SIREN-7 $\omega=60$ | Direct Residual | D-SDF | Ours |
|---|---|---|---|---|---|---|---|---|---|
| angel | 6.00/4.19 | 2.28/2.87 | 1.47/1.43 | 9.54/5.47 | 5.57/2.85 | -/- | 251/87.9 | 3.36/3.70 | **1.30/0.89** |
| asian dragon | 4.96/4.02 | 1.66/4.05 | 1.03/1.91 | 6.13/5.28 | 3.65/2.36 | 7.24/4.03 | 269/92.7 | 2.84/1.80 | **0.93/1.36** |
| camera | 4.62/1.51 | 1.56/1.15 | 1.32/0.62 | 6.50/2.38 | 4.11/1.10 | -/- | 281/38.5 | 1.99/2.01 | **1.25/0.34** |
| compressor | 5.55/1.35 | 1.64/0.88 | 1.52/0.44 | 8.83/2.82 | 4.63/0.82 | -/- | 330/56.9 | 2.66/3.71 | **1.39/0.23** |
| dragon | 5.10/4.00 | 1.80/3.77 | 1.39/2.37 | 7.04/4.75 | 3.80/2.49 | -/- | 263/76.4 | 2.68/**1.21** | **1.24**/1.50 |
| dragon warrior | 5.94/7.25 | 2.46/8.12 | 1.52/5.21 | 7.27/8.45 | 3.68/4.83 | 5.96/8.01 | 6.09/9.77 | 2.22/**4.46** | **1.47**/4.56 |
| dragon wing | 5.68/5.41 | 2.01/4.98 | 1.47/2.87 | 6.40/5.46 | 7.92/3.84 | -/- | 167/76.7 | 2.66/4.09 | **1.31/1.49** |
| dragon china | 6.20/2.31 | 2.32/1.75 | **1.39**/1.01 | 9.15/4.35 | 6.20/2.29 | -/- | 272/69.7 | 3.31/7.06 | 1.40/**0.51** |
| dragon cup | 4.39/3.13 | 1.83/3.57 | 1.24/1.17 | 7.67/4.69 | 5.50/2.15 | -/- | 173/86.9 | 3.21/4.93 | **1.10/0.51** |
| helmet | 4.79/1.02 | 1.70/0.871 | 1.40/0.410 | 8.40/2.72 | 5.19/0.83 | -/- | 263/94.6 | 2.59/1.99 | **1.29/0.13** |
| hunter | 4.17/4.66 | 2.03/5.08 | 1.18/2.08 | 8.96/6.66 | 3.40/2.58 | -/- | 3.39/3.64 | 2.61/4.89 | **0.91/1.14** |
| luyu | 7.22/4.29 | 2.19/3.30 | 1.53/1.81 | 8.98/6.83 | 6.16/3.16 | -/- | 206/94.6 | 5.10/9.76 | **1.28/1.02** |
| pearl dragon | 7.48/5.97 | 2.37/6.10 | 1.49/2.67 | 10.1/9.56 | 5.05/4.07 | -/- | 66.1 /49.8 | 3.26/6.24 | **1.30/1.43** |
| ramesses | 4.24/2.30 | 1.47/2.47 | 0.97/1.93 | 6.40/3.77 | 4.20/2.33 | -/- | 3.78/6.60 | 3.16/9.66 | **0.92/1.58** |
| Thai Statue | 5.23/7.01 | 7.16/16.7 | 1.30/4.77 | 6.27/7.48 | 3.81/4.17 | -/- | 117/45.2 | 1.76/**2.46** | **1.07**/2.92 |
| Vase Lion | 5.92/1.93 | 1.86/2.18 | 1.39/0.77 | 39.9/25.6 | 4.68/1.02 | -/- | 227/54.8 | 2.26/2.31 | **1.31/0.43** |
| AVG | 5.47/3.77 | 2.27/4.24 | 1.35/1.97 | 9.85/6.64 | 4.85/2.56 | -/6.02 | 181/59.0 | 2.85/4.39 | **1.22/1.25** |

Table 4: Detailed quantitative evaluation (corresponding to Table 1).

#### B.1.2  ANALYSIS

We provide further diagnosis for the underwhelming results from the direct residual and the D-SDF models. Direct residual composes the final SDF as a simple sum of the base SDF and a residual value. We train this model also with the attenuation and scaled $\tanh$ activation, and supervise both the base SDF and the composed SDF to stablize the base prediction. However, as shown in Figure 12a, the composed SDF often contain large structural errors. Further inspections show that during the training, such structural errors change quickly and the reconstruction oscillates in scale.

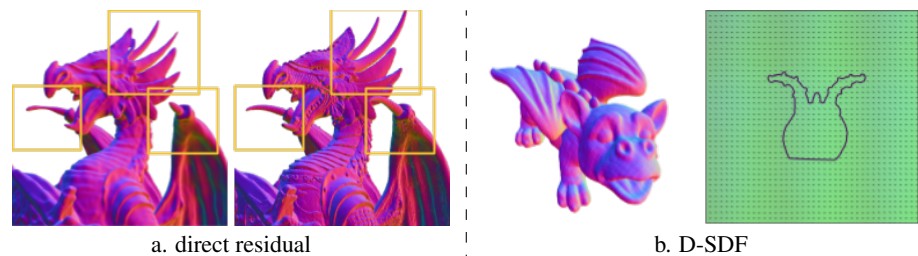

| a. direct residual | b. D-SDF |

Figure 12: Examples of the direct residual and D-SDF models.

Chamfer distance
points to point distance / normal cosine distance $\cdot 10^{-2}$

| training points | noise $\sigma$ | ours | poisson reconstruction |
|---|---|---|---|
| 40000 | 0.002 | 1.07/7.54 | 1.08/7.78 |
| 40000 | 0.005 | 1.05/7.57 | 1.08/7.82 |
| 400000 | 0.002 | 1.00/6.01 | 1.04/6.63 |
| 400000 | 0.005 | 1.00/5.99 | 1.04/6.60 |

Table 5: Quantitative evaluation given sparse and noisy inputs.

These indicate that the without the directional constraints on the displacement, the two networks are not sufficiently coupled and interfere with each other during training.

In D-SDF the displacement is not enforced to be along the normal direction. D-SDF yields competitive quantitative result, but as Figure 12b shows, the predicted displacement vectors are homogeneous, indicating that due to the lack of constraints the displacement network is not incentivised to return meaningful outputs.

### B.2 Discussion about $\omega_B$ and $\omega_D$

As pointed out in Section 3.2, $\omega$ controls the upper bound of the frequency the network is capable of representing. When using a single SIREN, a larger $\omega$ can represent higher frequencies (more details) but at the same time tends to create high-frequency artifacts and issues with convergence. Therefore, the choice of $\omega_B$ and $\omega_D$ is guided solely by two simple criteria: 1) $\omega_B$ should be relatively small to provide a smooth and stably trainable base surface. 2) $\omega_D$ should be sufficiently large to represent the amount of details observed in the given shape.

Based on the empirical experience of the previous work Sitzmann et al. (2020), for a baseline SIREN network, $\omega = 30$ provides a good balance for stability and detail representation. Based on this value, we choose $\omega_B = 15$, so that the base is smoother than the input surface, thereby creating a necessity for the displacement field; $\omega_D = 60$ is chosen empirically as a value that is capable of representing the high-frequency signals exhibited in the high-resolutions shapes we tested. If the input shape is very simple and smooth (*e.g.* the shape in the first row of Figure 13), the base SDF with $\omega_B = 15$ is already sufficient to represent the groundtruth surface, and the displacement has little impact. In order to enforce a frequency separation as in the detail transfer application, one can reduce $\omega_B$ (*e.g.* to 5, as shown in Figure 13(b)). For very detailed surfaces, $\omega_D$ needs to be high enough to enable sufficient resolution of the displacement field. We choose $\omega_D = 60$, which is suitable for all the tested shapes. When keeping $\omega_D$ fixed, varying $\omega_B$ determines the smoothness of the base, therefore also decides how much correction the displacement network must deliver. If $\omega_B$ is too small, the displacement network can become overburdened with the task, leading to faulty reconstruction and training instabilities.

### B.3 Stress tests

While we used dense and clean sampled point clouds as inputs in the paper, as our focus is on detail representation, we examine the behavior of our method under noisy and sparse inputs. Specifically, we train our network with 400 thousand and 40 thousand sampled points (10% and 1% of the amount in our main experiment, respectively), and added $\sigma = 0.002$ and $\sigma = 0.005$ Gaussian noise on both

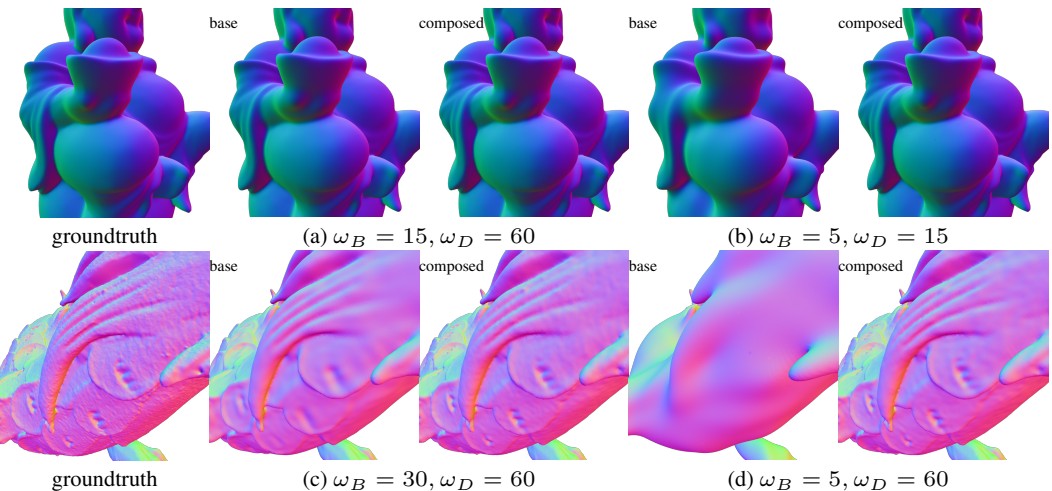

groundtruth      (a) $\omega_B = 15, \omega_D = 60$      (b) $\omega_B = 5, \omega_D = 15$

groundtruth      (c) $\omega_B = 30, \omega_D = 60$      (d) $\omega_B = 5, \omega_D = 60$

Figure 13: Effect of $\omega_B$ and $\omega_D$ demonstrated on meshes with different level of details. For smooth low-resolution meshes such as the example shown in the first row, a small $\omega$ suffices to represent all the details in the given mesh. In this case, the base SDF (*e.g.* $\omega_B = 15$) alone can accurately express the surface geometry, rendering the displacement network unnecessary, as a result the composed surface is indistinguishable from the base surface, as shown in (a). To enforce detail separation, one can reduce $\omega_B$, *e.g.* to 5, as shown in (b). On the other hand, given detailed meshes (such as in the second row), $\omega_D$ ought to be large enough to be able to capture the high-frequency signals. When keeping $\omega_D$ fixed, reducing $\omega_B$ increases the smoothness of the base SDF, as shown in (c) and (d), therefore also increases how much correction the displacement network must deliver. In the extreme case, $\omega_B = 0$, we would have a single high-frequency SIREN, which is subject to convergence issues, as shown in Tab 4.

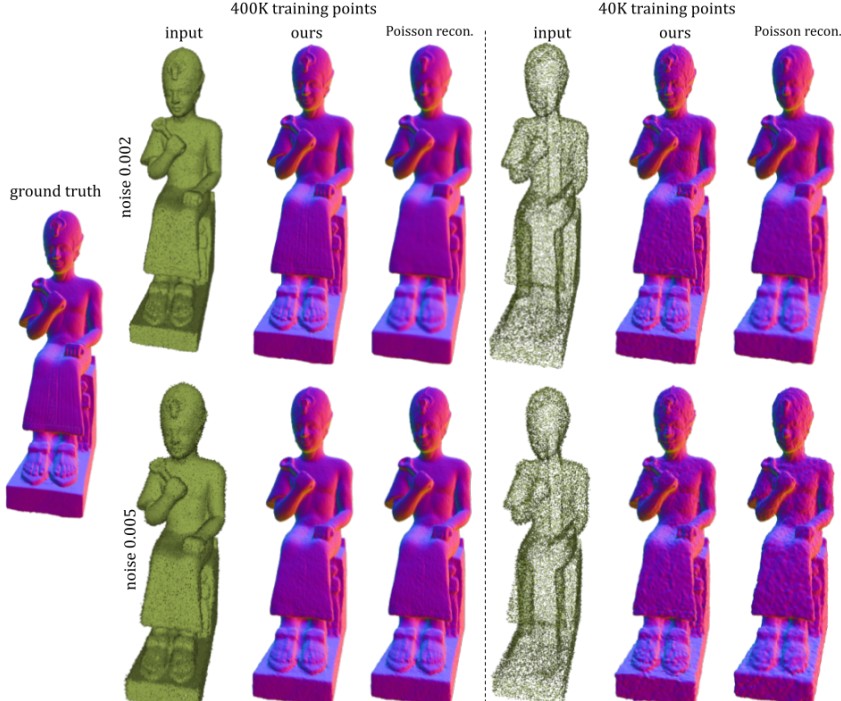

Figure 14: Qualitative evaluation given sparse and noisy inputs.

the point normals and the point positions. From the qualitative and quantitative results shown in Figure 14 and Table 5, we can see that our method recovers geometric details better than Poisson reconstruction given sufficient training data (c.f. the left half of the figure). When the training sample is sparse, our method tends to generate more high-frequency noise as a result of overfitting.

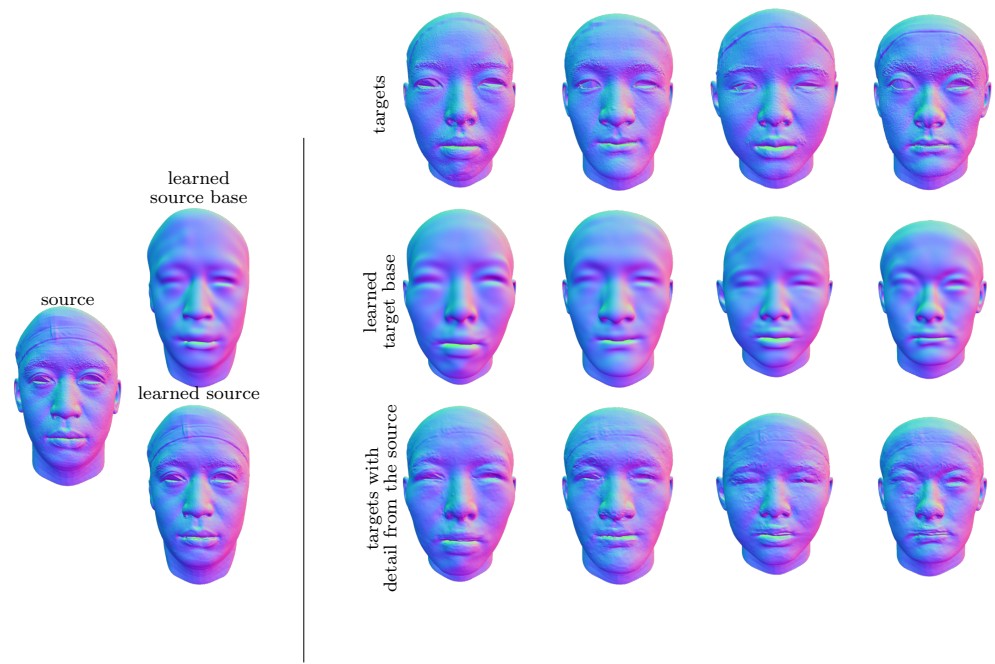

Figure 15: Detail transfer to multiple similarly aligned target shapes. Given a (detailed) source shape, we train a base network $\mathcal{N}^{\omega_B}$ to represent the smooth base surface (see *learned source base*), as well as a feature extractor $\phi$ and a transferrable displacement network $\mathcal{T}^{\omega_D}$ to represent the surface details. During detail transfer, we only need to fit the lightweight base network for each new target, while the feature extractor and displacement net can be applied to the new shapes without adaptation.

### B.4  DETAIL TRANSFER.

The proposed transferable IDF makes it possible to deploy $\phi$ and $\mathcal{T}^{\omega_D}$ a new base surface without any fine-tuning or adaptation. In other words, to transfer details to multiple targets, we only need to fit a new base network $\mathcal{N}^{\omega_B}_{\text{new}}$. The composed SDF $\hat{f}(\mathbf{x})$ can be computed simply by replacing $\mathcal{N}^{\omega_B}$ with $\mathcal{N}^{\omega_B}_{\text{new}}$. We show the result of a multi-target detail transfer in Figure 15. It is worth mentioning that even though the point extractor is trained on a single source shape, it is able to generalize across different identities thanks to the built-in scale and translation invariance.

### B.5  INFERENCE AND TRAINING TIME

Training the models as described in the paper takes 2412 seconds (around 40 minutes), which amounts to 120 epochs, i.e., around 20 seconds per epoch, where each epoch comprises 4 million surface samples and 4 million off-surface samples. In comparison, the original implementation of NGLOD6 takes 110 minutes to train 250 epochs, where each epoch comprises 200000 surface samples and 300000 off-surface samples. As for inference, using the same evaluation setup, NGLOD6 takes 193.9s for $512^3$ points, while our inference takes 250.06 seconds for $512^3$ query points. All benchmarking is performed on a single Nvidia 2080 RTX GPU. These timings could be improved by optimizing the model for performance, which we did not. For example, instead of using autodiff for computing the base surface normals, one could exploit the fact that the differentiation of SIREN is also a SIREN and explicitly construct a computation graph for computing the base surface normals.

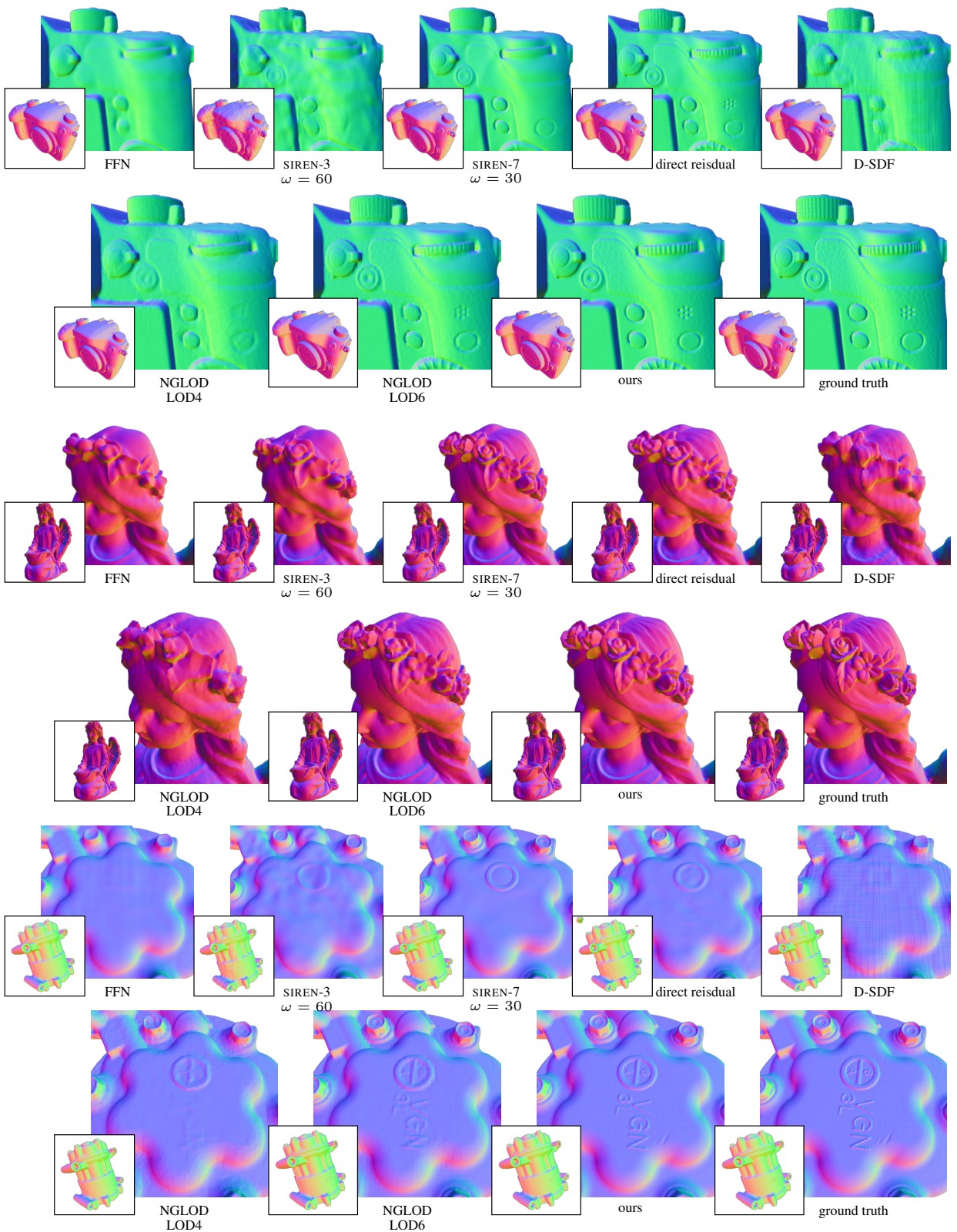

Figure 16: Comparison of detail reconstruction (better viewed with zoom-in). Methods that did not converge are omitted in the visual comparison.

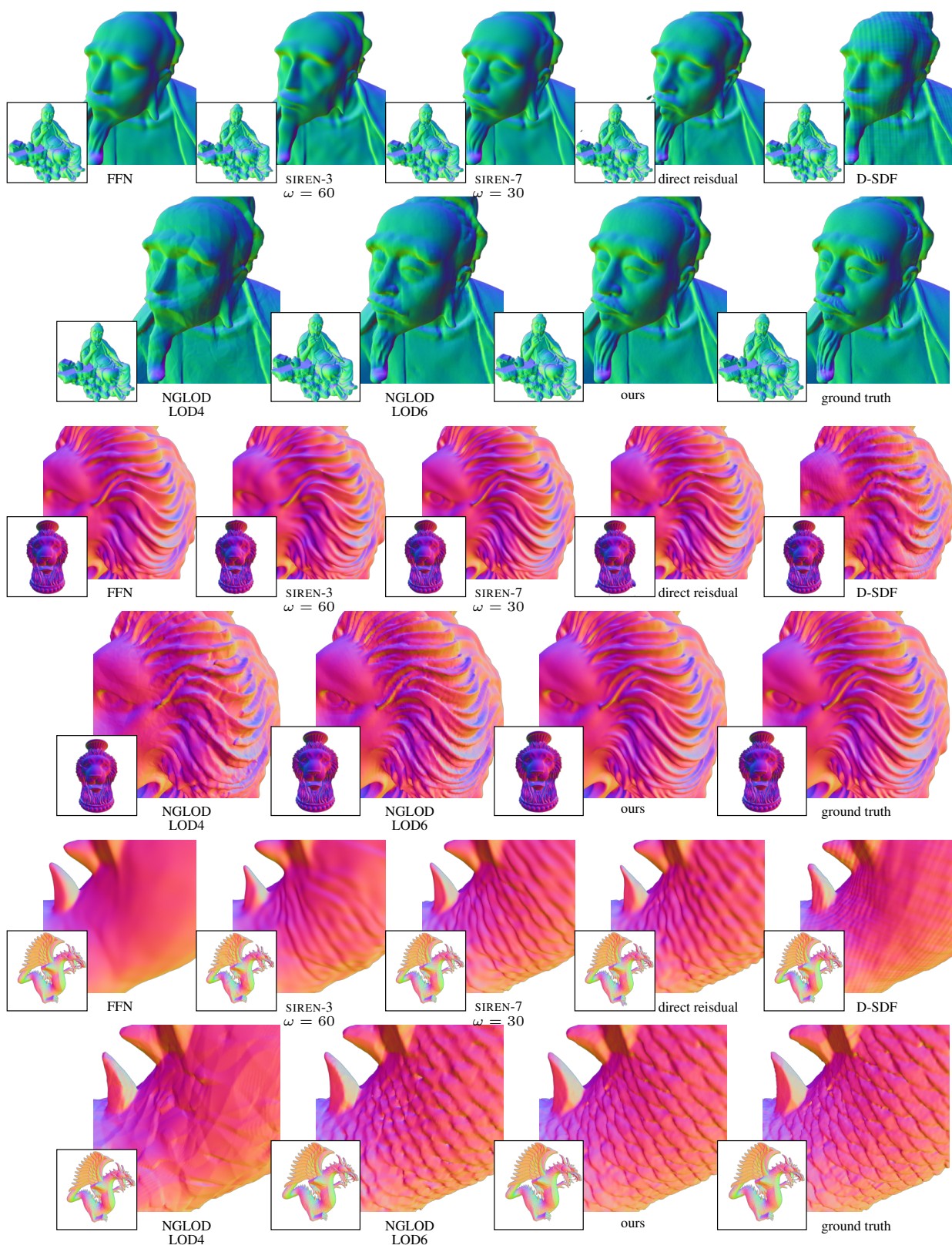

Figure 16: (Cont.) Comparison of detail reconstruction (better viewed with zoom-in). Methods that did not converge are omitted in the visual comparison.

