# OpenReview forum: "Geometry-Consistent Neural Shape Representation with Implicit Displacement Fields"
_ICLR.cc/2022/Conference — ICLR 2022 Poster_

### Official Review · Reviewer_WHEF · 2021-10-29

**Correctness:** 4
**Technical Novelty And Significance:** 3
**Empirical Novelty And Significance:** 4
**Recommendation:** 10
**Confidence:** 4

**Main Review:**

he paper is well written, makes its contribution very clear and gives a distinguished overview about related work. I had no problems following the text, and the equations are

The papers main contribution is the introduction of a two-level implicit representation (both based on SIREN) for 3D geometry which works exceptionally well. The general idea was already introduced for single-view reconstruction (in Li & Zhang as mentioned by the authors), but this papers evolves the setting to work for general geometry representation, including an inverse displacement and making the displacement transferrable by using a kind of feature-based UV map.

While the existing two SIREN network setup lowers the value of the contribution, the elegant solution for some geometry representation specific problems. These solutions are the expensive comparison to the ground-truth which is solved by including an inverse implicit field, and learning a feature based UV field (computing consistent UV maps on different surfaces is very hard). I could see especially the second part being reused in other applications as they allow to transfer the UV to similar shapes.

The results are visually impressive, especially the texture transfer, closer to the ground-truth than all competitors all while using a smaller model size. The results are both smooth in areas where this is correct, as well as able to show details where necessary.  This could be incredible useful for applications based on implicit 3D representations that struggle with training complexity of highly detailed models.

The ablation study is well done, however the quantitative comparison is only done on a single dataset, SketchFab-16. I do not know this database and the provided link only goes to a general website with a wide collection of 3D models (some of them not free). This makes it impossible to exactly reproduce the results. Additionally, it seems all models on this page are of a certain high quality, it would have been interesting to at least see some other examples with shapes of different quality.

Minor comments:
- I find that the practice of the listed citations without brackets around them very distracting when reading. I do not know if this is part of the conference style guide but I would recommend to change it.
- Figure 4: It would be great to write the w parameter directly under/above each image.

**Summary Of The Paper:**

The paper proposes a an implicit representation of surface displacement fields that is well learnable by neural networks. The network is based on two SIREN architectures that separate low and high frequency information in the model. One returns an SDF of the rough shape while the second returns a displacement field at the surface of the SDF. In addition to this architecure (which existed before but for a different application), the paper introduces an inverse implicit mapping that speeds up the evaluation of the loss, and a feature based UV map that can be transferred between similar shapes. The paper compares to previous work on SketchFab, and specifically evaluates against a vanilla SIREN model where this method results in both a smaller model size as well as more detailed reconstruction. Additionally, there are experiments that show how details can be transferred across different (but similar) models.

**Summary Of The Review:**

While neither the SIREN architecutre nor the displacement field is a novel contribution, their combination is not trivial and shows exceptional results in terms of quality and model size. In addition, the paper presents some elegant solutions for efficiently evaluating the loss in this new setting, and using a feature based UV mapping that can be transferred to similar shapes.
I think these are sufficient for publication. The quantitative evaluation is a bit lacking, and the final version of this paper should include an exact list of shapes that were used in the evaluation. op

---

> ### Author Response · Authors · 2021-11-22
> **response**
>
> Thank you for your positive inputs!
>
> Regarding the meshes used in the model, our plan is to release all 16 benchmarked shapes together with the code. We think this would be the most straightforward way for others to test our method.
>
> Regarding the variation of resolution in the shapes, since our purpose is detail reconstruction, we aim at finding the highest-quality meshes under the given licensing. The meshes we included have 1.6 million to 10 million faces. Nonetheless, the discussion and additional experiments (Section B.2) that we added in the Supplementary material may address some aspects of this question.

---

> > ### Comment · Reviewer_WHEF · 2021-11-29
> > **final response**
> >
> > Thank you for the clarification, that answers all my questions. 10 million faces is quite impressive.

---

### Official Review · Reviewer_Eb6T · 2021-11-02

**Correctness:** 4
**Technical Novelty And Significance:** 3
**Empirical Novelty And Significance:** 3
**Recommendation:** 6
**Confidence:** 3

**Details Of Ethics Concerns:**

The proposed method can represent 3D shapes with details. Its transferability of IDF also enables shapes manipulation. Care should be taken in applying such a technique.

**Main Review:**

Strengths

(1) To represent both the coarse surfaces and displacements implicitly by two neural networks is new. The results show that the proposed method can keep detailed 3D geometry while remaining a reasonably small network size.

(2) Instead of naively building the displacement SDF as residuals, the proposed displacement field and attenuation function (eq(1),eq(3)) are interesting. Ablation studies also demonstrate their effectiveness.

Weaknesses

(1) I would like to see more experiments and discussions about how the parameters ωB and ωD (ωB = 15 and ωD = 60) are chosen in the paper. How would they affect the performance in different objects, from simple to complex surfaces? To me, it seems that these two parameters would be object-dependent.

(2) Is this method generalizable to other baseline models (other than SIRENs)?

**Summary Of The Paper:**

This paper proposes an implicit neural representation for 3D geometry, named implicit displacement fields. The method is consists of two SIREN networks as baseline models to approximate: a coarse base shape representation (low-frequency) and an implicit displacement field (for high-frequency details). Different from previous works, the displacement field is implicitly represented by a SIREN network. The coarse and fine networks design also enables applications such as shape manipulation.

**Summary Of The Review:**

The proposed implicit displacement fields have the ability to represent detailed geometries in relatively small network sizes. Experiments are done thoroughly to demonstrate the effectiveness of each component of the method.  If shapes are pre-aligned properly, the implicit displacement also enables the shape manipulation application. Despite having some uncertainties (see weaknesses), the reviewer holds a positive opinion on this paper.

---

> ### Author Response · Authors · 2021-11-22
> **response**
>
> Thank you for your feedback!
>
> > I would like to see more experiments and discussions about how the parameters ωB and ωD (ωB = 15 and ωD = 60) are chosen in the paper. How would they affect the performance in different objects, from simple to complex surfaces? To me, it seems that these two parameters would be object-dependent.
>
> As pointed out in Section 3.2, $\omega$ controls the upper bound of the frequency the network is capable of representing. When using a single SIREN, a larger $\omega$ can represent higher frequencies (more details) but at the same time tends to create high-frequency artifacts and issues with convergence. Therefore, the choice of  $\omega_B$ and $\omega_D$ is guided solely by two simple criteria:
> 1. $\omega_B$ should be relatively small to provide a smooth and stably trainable base surface.
> 2. $\omega_D$ should be sufficiently large to represent the amount of details observed in the given shape.
>
> If the input shape is very simple and smooth, the base SDF with $\omega_B=15$ is already sufficient to represent the groundtruth surface, thus the displacement has little impact. In this case, one can reduce $\omega_B$ in order to enforce a frequency separation, as in the detail transfer application. For very detailed surfaces, $\omega_D$ needs to be high enough to enable sufficient resolution of the displacement field. When keeping $\omega_D$ fixed, varying $\omega_B$ determines the smoothness of the base, therefore also decides how much correction the displacement network must deliver. If $\omega_B$ is too small, the displacement network can become overburdened, leading to faulty reconstruction and training instabilities. In the extreme case, $\omega_{B}=0$, we would have a single high-frequency SIREN, which is subject to convergence issues, as we showed in Table 4.
>
> Based on the empirical experience of the original SIREN paper, for the baseline SIREN model  $\omega=30$ provides a good balance for stability and detail representation. Based on this value, we choose $\omega_B=15$, so that the base is smoother than the input surface, thereby creating a necessity for the displacement field; $\omega_D=60$ is chosen empirically as a value capable of representing the high-frequency signals exhibited in all the high-resolution shapes we tested.
>
> *We added this discussion with concrete experiments in the supplementary material.*
>
> > Is this method generalizable to other baseline models (other than SIRENs)?
>
> As explained in Section 3.2 paragraph 2, our method builds upon SIREN’s controllable inductive bias for the output signal, which is the key to separating two frequency levels without explicit supervision or regularization. Hence we do not think it is feasible nor meaningful to use other baseline models.

---

### Official Review · Reviewer_GmTU · 2021-11-02

**Correctness:** 4
**Technical Novelty And Significance:** 3
**Empirical Novelty And Significance:** 3
**Recommendation:** 6
**Confidence:** 4

**Main Review:**

Strengths
- The paper is well-written and well-motivated. The authors have done adequate derivation from the classic explicit displacement mapping method, making the proposed method theoretically sound.
- The idea of implicitly representing both base surface and displacement map with neural networks is new and elegant. Transferable implicit displacement field (IDF) also adds to the novelty of this framework.
- The proposed method outperforms all baseline methods both quantitatively and qualitatively. The authors have done adequate ablation studies to validate the method design. The transfer experiments are impressive and show great potential in applying the method to downstream shape processing tasks such as detail modeling.

Weakness
- It would be helpful to explain how some of the hyper-parameters are chosen e.g. attenuation factors $\nu$, and frequency factors $\omega_B, \omega_D$

Other
- Is it possible to perform hierarchical displacement mapping / shape modeling based on this idea? Any discussion is welcomed.



**Summary Of The Paper:**

This paper proposes an implicit representation for shapes, which utilizes two SIREN networks of different frequencies representing base surface and displacement respectively. In addition, the authors introduces transferable implicit displacement fields, which enables the transferability of displacements between aligned shapes. The method is a natural extension to the displacement mapping method and is theoretically grounded. The evaluation results show that this method could represent shapes with sharper details while using less storage compared to baseline methods.

**Summary Of The Review:**

The paper proposed a new method for implicit shape representation. The authors have done adequate experiments to show the effectiveness and transferability of the proposed method. However the method is an intuitive combination of traditional displacement mapping method and SIREN, lowering the novelty of this method slightly.

---

> ### Author Response · Authors · 2021-11-22
> **response**
>
> Thank you for your feedback! We address the questions below:
>
> > It would be helpful to explain how some of the hyper-parameters are chosen e.g. attenuation factors $\nu$, and frequency factors $\omega_B$, $\omega_D$.
>
> We introduced $\nu$ in Section 3.2 and ablated its influence in Section 4.2 (Table 2 and Table 3). In summary, changes in $\nu$ marginally affect the model’s performance; a slight performance drop can be observed when $\nu$ is too large, e.g. $\nu>=0.2$ (10 times of the value set in the experiments in the paper).
>
> As for the $\omega$s, as pointed out in Section 3.2, $\omega$ controls the upper bound of the frequency the network is capable of representing. When using a single SIREN, a larger $\omega$ can represent higher frequencies (more details) but at the same time tends to create high-frequency artifacts and issues with convergence. Therefore, the choice of  $\omega_B$ and $\omega_D$ is guided solely by two simple criteria:
> 1. $\omega_B$ should be relatively small to provide a smooth and stably trainable base surface.
> 2. $\omega_D$ should be sufficiently large to represent the amount of details observed in the given shape.
>
> If the input shape is very simple and smooth, the base SDF with $\omega_B=15$ is already sufficient to represent the groundtruth surface, thus the displacement has little impact. In this case, one can reduce $\omega_B$ in order to enforce a frequency separation, as in the detail transfer application. For very detailed surfaces, $\omega_D$ needs to be high enough to enable sufficient resolution of the displacement field. When keeping $\omega_D$ fixed, varying $\omega_B$ determines the smoothness of the base, therefore also decides how much correction the displacement network must deliver. If $\omega_B$ is too small, the displacement network can become overburdened, leading to faulty reconstruction and training instabilities. In the extreme case, $\omega_{B}=0$, we would have a single high-frequency SIREN, which is subject to convergence issues, as we showed in Table 4.
>
> Based on the empirical experience of the original SIREN paper, for the baseline SIREN model  $\omega=30$ provides a good balance for stability and detail representation. Based on this value, we choose $\omega_B=15$, so that the base is smoother than the input surface, thereby creating a necessity for the displacement field; $\omega_D=60$ is chosen empirically as a value capable of representing the high-frequency signals exhibited in all the high-resolution shapes we tested.
>
> *We added this discussion with concrete experiments in the supplementary material.*
>
> > Is it possible to perform hierarchical displacement mapping / shape modeling based on this idea? Any discussion is welcomed.
>
> We think IDF provides a solid foundation for exploration in this direction. For example, one can extend the current framework to have more than one displacement network. Here, the research focus would be optimizations to the current implementation to avoid higher-order differentiation. On the other hand, one may also explore global-local context for the displacement, i.e. apply displacement gradually in a finer scale. Such an approach may make the framework generalizable to collections of shapes.

---

### Official Review · Reviewer_G54X · 2021-11-04

**Correctness:** 4
**Technical Novelty And Significance:** 2
**Empirical Novelty And Significance:** 3
**Recommendation:** 5
**Confidence:** 4

**Main Review:**

Strengths:

S1: The paper presents an interesting idea. Although this is a straightforward idea, using implicit displacement fields is up to my knowledge new.

S2: Related work is OK, and the paper is well organized.

S3: Several experimental results are presented, including ablation studies. However, essential comparisons are missing (see the weaknesses.)

Weaknesses:

W2: The method is mostly constructed on top of previous methods; there are no network changes or losses. There is a contribution in the signed distance function and a pipeline for transferable implicit displacement fields. Why are we using two SIRENs for f and d? Shouldn't the d be a simpler network?

W3: Considering the experimental result. I feel that the method will fail with noise because of things like the need for normal to the points. This is a very relevant fact, and it is not described in the document. There are small tests with noise in the appendices, but the level of noise added is almost 0.

**Summary Of The Paper:**

The paper addresses the problem of obtaining a 3D surface reconstruction from point clouds. This has been one of the most studied topics in 3D vision, with different kinds of approaches. The authors tackle this problem using implicit representations, specifically using signal distance functions and query features. The main contributions are a consideration of implicit displacements for modeling different frequency levels and introducing a transferable implicit displacement field that replaces the common coordinates inputs with constructed transferable features.

**Summary Of The Review:**

Although the proposed method builds mainly on top of previous techniques, the paper presents some interesting contributions. However, it is not clear how this new representation will work with noise, and I believe it would be beneficial to clarify this. However, if the other reviewers think the paper is ready to be accepted, I will not be against it.

---

> ### Author Response · Authors · 2021-11-22
> **response**
>
> Thank you for your input. Based on your review, we think it is necessary to recapitulate a few important properties of our method, which might have been missed.
> - Our method is the first to consider spectral shape representation in the context of implicit shape representation and the first to represent displacement fields by implicit functions. We believe these are important theoretical contributions to the highly active study of implicit representations.
> - Regarding network architecture, we propose a novel network architecture that non-trivially combines two SIRENs to achieve frequency decomposition by leveraging their distinct inductive biases with respect to signal frequency. In addition, we also introduce novel components to make the learned high-frequency details transferable. We believe these are significant architecture changes.
> Note that using SIRENs is necessary due to their special properties in the tunable inductive bias of signal frequency.
> - Regarding losses, we put extra effort into the theoretical formulation (Section 3.1) to allow efficient computation of SIREN’s initially proposed training loss, which leverages the point normals. However, our model can be trained without this information too at the cost of requiring the groundtruth SDF value at every query point -- as in many other methods, which is inarguably challenging to obtain accurately. While it is true that normals are more sensitive to noise, we have shown the robustness of our method in Appendix B.2, where the noise injected is at a reasonable level. Moreover, our method mainly focuses on an accurate and efficient representation of highly detailed shapes; tolerance to high noise levels is out of the scope of this paper.
>
> ---
> Below we address some specific comments in your review:
>
> > Nevertheless, the paper has some interesting ideas, and I would like to recommend the authors pursue these. Although many comparisons are missing, the results are promising.
>
> We respectfully ask for a more specific description about which interesting ideas we should pursue more and what comparisons are missing.
>
> > There are many methods listed in the literature reviews, the current state of the art methods, that are not tested.
>
> We respectfully ask the reviewer to provide a specific list of other methods that we should compare with. If relevant and feasible, we will gladly evaluate them.

---

### Decision · Program_Chairs · 2022-01-20

**Decision:**

Accept (Poster)

**Comment:**

This paper provides a normal map-inspired implicit surface representation involving a smooth surface whose high frequency detail comes from normal displacements.  Reviewers were impressed with the results and theoretical discussion in the paper.  The AC agrees.

The authors were responsive to reviewer feedback and addressed some questions about parameter choice during the rebuttal phase, including new experiments/discussion in the supplementary document.  Note the response to reviewer WHEF notes that the authors will be releasing data/code; the AC strongly hopes the authors are true to their word in that regard.

The AC chose to disregard some comments from reviewer G54X regarding tests with noise, as this method appears to be tuned to computer graphics applications; the level of empirical work here aligns with past work in the area.  Of course the authors are encouraged to include some tests responding to the reviewer comments in the camera ready.  The AC also found the score from reviewer WHEF to be somewhat uncalibrated with the tone of their review, but of course their assessment is quite positive nonetheless.


One small comment:  The abstract appears a bit strangely on the OpenReview site because of line breaks; if possible, please remove the line breaks.

Another small comment:  The "spectral shape representation" phrase used a bit in the discussion below might not be advisable, as this phrase typically refers to the intrinsic spectrum of a shape (e.g. Laplace-Beltrami analysis)